# Atmospheric Distribution of HCN from Satellite Observations and 3-D Model Simulations

Antonio G. Bruno[1,2], Jeremy J. Harrison[1,2], Martyn P. Chipperfield[3,4], David P. Moore[1,2], Richard J. Pope[3,4], Christopher Wilson[3,4], Emmanuel Mahieu[5], and Justus Notholt[6]

[1]School of Physics and Astronomy, University of Leicester, Leicester, UK
[2]National Centre for Earth Observation, University of Leicester, Leicester, UK
[3]School of Earth and Environment, University of Leeds, Leeds, UK
[4]National Centre for Earth Observation, University of Leeds, Leeds, UK
[5]Institut d'Astrophysique et de Géophysique, Université de Liège, Liège, Belgium
[6]Institute of Environmental Physics, University of Bremen, Bremen 28334, Germany

**Correspondence:** A. G. Bruno (agb22@leicester.ac.uk)

**Abstract.**

Hydrogen cyanide (HCN) is an important tracer of biomass burning but there are significant uncertainties in its atmospheric budget, especially its photochemical and ocean sinks. Here we use a tracer version of the TOMCAT global 3-D chemical transport model to investigate the physical and chemical processes driving the abundance of HCN in the troposphere and stratosphere over the period 2004-2016. The modelled HCN distribution is compared with version 4.1 of the Atmospheric Chemistry Experiment - Fourier Transform Spectrometer (ACE-FTS) HCN satellite data, which provides profiles up to around 42 km, and with ground-based column measurements from the Network for the Detection of Atmospheric Composition Change (NDACC). The long-term ACE-FTS measurements reveal the strong enhancements in upper tropospheric HCN due to large wildfire events in Indonesia in 2006 and 2015. Our 3-D model simulations confirm previous lower altitude balloon comparisons that the currently recommended NASA Jet Propulsion Laboratory (JPL) reaction rate coefficient of HCN with OH greatly overestimates the HCN loss. The use of the rate coefficient proposed by Kleinböhl et al. (2006) in combination with the HCN oxidation by $O(^1D)$ gives good agreement between ACE-FTS observations and the model. Furthermore, the model photochemical loss terms show that the reduction of the HCN mixing ratio with height in the middle stratosphere is mainly driven by the $O(^1D)$ sink with only a small contribution from reaction with OH. From comparisons of the model tracers with ground-based HCN observations we test the magnitude of the ocean sink in two different published schemes (Li et al., 2000, 2003). We find that in our 3-D model the two schemes produce HCN abundances which are very different to the NDACC observations but in different directions. A model HCN tracer using the Li et al. (2000) scheme overestimates the HCN concentration by almost a factor two, while a HCN tracer using the Li et al. (2003) scheme underestimates the observations by about one-third. To obtain good agreement between the model and observations we need to scale the magnitudes of the global ocean sinks by factors of 0.25 and 2 for the schemes of Li et al. (2000) and Li et al. (2003), respectively. This work shows that the atmospheric photochemical sinks of HCN now appear well constrained but improvements are needed in parameterising the major ocean uptake sink.

## Plain Language Summary

Hydrogen cyanide (HCN) is one of the most abundant cyanides present in the global atmosphere. It is mainly emitted into the atmosphere from biomass burning with minor contributions from anthropogenic sources. Uptake by the ocean provides the main tropospheric sink while in the stratosphere HCN is removed via photochemical reactions. Using the TOMCAT three-dimensional atmospheric chemical transport model, satellite data and ground-based observations, we have investigated the physical and chemical processes driving HCN variability. In particular, we present HCN profile observations from the ACE-FTS instrument which extend up to around 42 km. We find that the HCN oxidation by $O(^1D)$ drives the HCN loss in the middle stratosphere. The currently recommended OH reaction rate is also confirmed to greatly overestimate the HCN atmospheric loss while the rate proposed by Kleinböhl et al. (2006) gives a better agreement with profiles measured by ACE-FTS. We also evaluated two ocean uptake schemes proposed by Li et al. (2000, 2003) using our model and found them to be unrealistic in reproducing the HCN ocean sink. Due to the large differences between the tracers using the default ocean uptake schemes, we need to scale these schemes in different ways to obtain good agreement with HCN observations.

## 1 Introduction

Hydrogen cyanide (HCN) is one of the most abundant cyanides in the atmosphere (Singh et al., 2003), and can influence the nitrogen cycle and reactive nitrogen ($NO_x$) (Li et al., 2000, 2003, 2009). Previous modelling studies (Li et al., 2000, 2003, 2009; Singh et al., 2003; Kleinböhl et al., 2006) have shown that the HCN variability is mainly determined by biomass burning, as the dominant source, and by ocean uptake, as the major tropospheric sink. HCN is, in fact, released into the atmosphere predominantly by biomass burning events with only a minor contribution from industrial activities. In addition, Li et al. (2000, 2003) suggest that ocean uptake is the major loss process at the surface with a non-negligible contribution from the oxidation by OH radicals in the troposphere (Cicerone and Zellner, 1983). As a result, the HCN tropospheric lifetime is estimated at about 5 months (Singh et al., 2003; Li et al., 2000, 2003). The stratospheric HCN reduction instead is caused by photochemical loss, with a resulting 4-5 years lifetime in the stratosphere (Cicerone and Zellner, 1983; Li et al., 2000, 2003). Due to its low chemical reactivity and long lifetime, HCN is a good atmospheric tracer of biomass burning events. However, the atmospheric HCN budget and the processes driving its variability are still not fully understood (Li et al., 2003; Singh et al., 2003). The rate constant for the reaction between HCN and OH and the rate at which HCN is reduced by ocean uptake still have a number of significant uncertainties. The present study aims to clarify some of the key uncertain aspects in HCN sources and sinks using 3-D model simulations, satellite solar occultation data and ground-based measurements of HCN in the atmosphere.

HCN column measurements from ground-based Fourier transform infrared (FTIR) spectrometers are available at several sites from the Network for the Detection of Atmospheric Composition Change (NDACC) (De Mazière et al., 2018). These measurements are an important source of information on the spatial and temporal distribution of HCN, but are too sparse to represent a strong global constraint on HCN emissions and removal processes. HCN volume mixing ratio (VMR) profiles measured during balloon-borne campaigns provide more accurate information on the vertical distribution of HCN (Kleinböhl et al., 2006) but with limitations in spatial and temporal coverage. Some limitations of the balloon observations are overcome

by satellite observations which offer global coverage and can place the balloon profiles measured at a few locations into a wider context, also extending the sensing range to the stratosphere. Together, this information provides a good constraint on the HCN concentrations and processes driving HCN variability on a global scale.

The Atmospheric Chemistry Experiment Fourier Transform Spectrometer (ACE-FTS), launched in 2004, was one of the first satellite instruments to measure HCN VMRs in the lower stratosphere. ACE-FTS measures HCN VMRs from the mid-troposphere up to ∼42 km with ∼3 km vertical resolution (Boone et al., 2005, 2020; Sheese et al., 2017), the extended altitude range allows us to test the HCN stratospheric loss, not otherwise possible with balloon-borne missions. The Microwave Limb Sounder (MLS) on the Aura satellite has also been measuring HCN mixing ratios since its launch in 2004. Both satellites have been used, frequently in combination, to study HCN variability in the upper troposphere and lower stratosphere (UTLS) and the biomass burning emission impact on the HCN concentrations (Pumphrey et al., 2006, 2018; Li et al., 2009; Sheese et al., 2017; Park et al., 2021). The MLS v5.0 data product for HCN has extremely large systematic errors in the lower stratosphere. For this reason, the data are not recommended for scientific use outside the upper stratosphere at pressures greater than 21 hPa (altitudes below ∼27 km) (Livesey et al., 2022), so we decided to perform our study using only ACE-FTS data.

Atmospheric chemical transport models (CTMs) are excellent tools for testing our understanding of atmospheric processes using a variety of observations and for deriving global tracer budgets. Here we use a tracer version of the detailed TOMCAT 3-dimensional CTM adapted to include the processes driving HCN variability (Bruno et al., 2022). TOMCAT is used in order to quantify the role played by the different stratospheric HCN loss mechanisms in determining the HCN variability observed from satellite measurements, and to assess how well currently available parameterisations perform in simulating HCN. Two rate coefficients for HCN oxidation by OH radicals have been compared, one based on the current JPL recommendation, which was last revised in 1983 (Burkholder et al., 2015, 2019), and the other from Kleinböhl et al. (2006) and Strekowski (2001). We show that the JPL recommended rate largely overestimates the HCN loss, while the use of the other Kleinböhl et al. (2006) rate coefficient significantly improves the agreement between the measurements and the model. ACE-FTS version 4.1 HCN data (Bernath et al., 2021) have been used to validate the modelled HCN distribution over the years 2004-2016. The model tracers have also been used to understand the ocean uptake contribution in the tropospheric HCN variability. Two ocean uptake fluxes from Li et al. (2000) and Li et al. (2003) were added into the TOMCAT model and evaluated using ground-based FTIR HCN measurements from the NDACC network.

## 2 HCN Measurements

### 2.1 Satellite Observations

The Atmospheric Chemistry Experiment Fourier Transform Spectrometer (ACE-FTS) is a high spectral resolution (0.02 cm$^{-1}$) limb sounder instrument on board the Canadian Science Satellite mission (SCISAT). SCISAT moves along a circular low earth orbit at an altitude of 650 km with an inclination of 74° and an orbit period of 97.7 minutes (Bernath et al., 2005). This orbit gives a sampling pattern with a high density of observations at high latitudes. The original aim of the SCISAT mission was to obtain a complete insight into the physical and chemical processes driving the distribution of ozone by measuring changes in

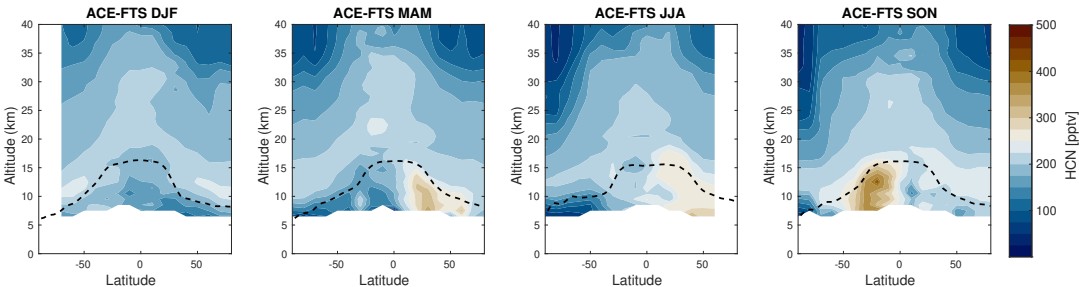

**Figure 1.** Latitude-height cross sections of ACE-FTS HCN zonal mean profiles (pptv) in $10°$ latitude bins for 4 seasons, December-February (DJF), March-May (MAM), June-August (JJA) and September-November (SON), from December 2008 to November 2009. The black dashed lines show the season-averaged location of the thermal tropopause (Maddox and Mullendore, 2018) based on ECMWF ERA-Interim reanalyses for the same periods.

atmospheric composition, in particular in polar regions. ACE-FTS operates in the infrared absorption spectrum over the range
750-4400 cm$^{-1}$ (2.2 to 13.3 $\mu$m) using the solar occultation technique, measuring a wide range of molecules in the upper troposphere and stratosphere (Bernath et al., 2005). Its profiles have been widely used as a reference for interpreting ground-based and nadir satellite measurements of HCN and to help validate tropospheric-stratospheric transport in atmospheric models (Park et al., 2013; Viatte et al., 2014; Glatthor et al., 2015; Sheese et al., 2017). The retrieved HCN profiles extend from the middle troposphere ($\sim$6-8 km) to $\sim$42 km with a vertical resolution on the order of $\sim$3 km (Boone et al., 2005, 2020). ACE-
FTS observations extend higher than the profiles measured by balloon-borne missions (Kleinböhl et al., 2006), allowing us to investigate stratospheric HCN variability and to test the different processes driving HCN loss. Here we present ACE-FTS version 4.1 HCN data (Bernath et al., 2021), the most recent update version, and use it to evaluate different modelled HCN tracers.

Figure 1 shows the seasonal ACE-FTS of the sample year 2008–2009 HCN zonal mean cross sections in $10°$ latitude bins. To
highlight the HCN stratospheric distribution, the black dashed line shows the seasonal average location of the tropopause, based on the WMO thermal tropopause definition (Maddox and Mullendore, 2018) using ECMWF ERA-Interim reanalyses (Dee et al., 2011). The first panel of Figure 1 shows the December–February (DJF) season which exhibits a high upper tropospheric HCN concentration in the Southern Hemisphere from mid- to high latitudes. In the March-May (MAM) season an enhancement of upper tropospheric HCN is observed in the Northern Hemisphere (NH) from the tropics to midlatitudes, while in the June-
August (JJA) season the enhancement is observed in the NH mid- to high latitudes. The observed behaviour is attributable to biomass burning emissions during the wildfire seasons in the tropical and midlatitude regions, respectively. During the September-November (SON) season an enhancement of HCN in the upper troposphere is observed over southern tropics, due to biomass burning emissions from South America, Africa and South East Asia.

Figure 2 shows the time-altitude cross sections of the two-month mean HCN mixing ratios measured by ACE-FTS averaged
over three $30°$ latitude bands. Here, the strong HCN tropospheric seasonal variability is clearly visible, and is followed by a

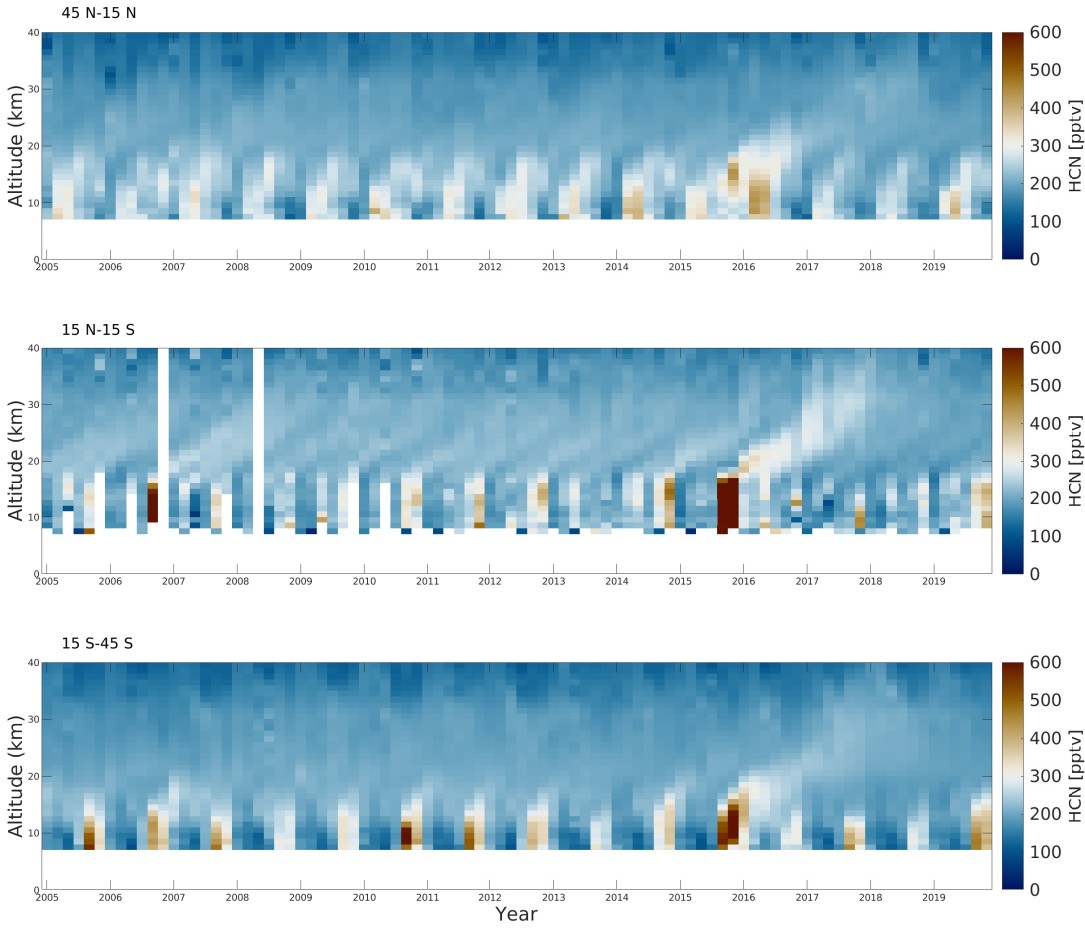

**Figure 2.** Time-altitude sections of two-month-mean HCN mixing ratios (pptv) from ACE-FTS averaged into three 30° latitude bins, 45°N-15°N, 15°N-15°S and 15°S-45°S for 2005–2020. Note that the colour scale saturates and the maximum observed HCN mixing ratio in the tropical band in 2015 is over 1300 pptv.

similar seasonal cycle in the stratosphere at all latitudes, in agreement with Park et al. (2021). During the months following the period of high tropospheric HCN concentration, a large amount of HCN is transported to the stratosphere, where it persists for the following years. In particular, after the El Niño events, which influence the fire season of South East Asia, this amount is notably high. Looking specifically at the tropical region, two of the largest El Niño events ever recorded in Indonesia, in 2006 and 2015, are highlighted by an extremely high concentration of HCN emitted during peat fires. During the months following the two events, a large quantity of HCN was transported to the stratosphere and persisted for a longer time than the typical seasonal variation. Specifically, after the 2015 Indonesia fire season, the HCN transported to the stratosphere persisted for the following two years before being completely reduced. Sheese et al. (2017) observed the transport of the HCN from the

upper troposphere to the lower stratosphere during January and February 2016 and its persistence during the entire year using
ACE-FTS measurements. This behaviour is also visible in the latitude band 15°S-45°S.

## 2.2 Ground-based FTIR Observations

NDACC is an international global network of more than 90 ground-based stations that measure atmospheric composition in order to detect long-term changes and trends in the chemical and physical state of the atmosphere (De Mazière et al., 2018). The stations employ a variety of techniques and instruments, and observations at some sites extend from the early 1990s. Because of the different instruments and station set-ups, not all species are available at all stations. The Fourier Transform Infra-Red (FTIR) instruments are used in around 25 NDACC sites. They consist of a high quality FTIR spectrometer combined with a high precision solar tracker, which records the direct solar absorption spectra in the mid-infrared spectral region. The instruments work only under clear sky conditions, i.e. the instrument view must be free of clouds and no measurements are possible during the night, including, of course, polar night. This impacts the ground-based FTIR sampling frequency, with typically an average of 120 observation days per year, when considering all sites. Note, however, that for some stations, yearly coverage can be significantly larger, lying regularly in the range 240-270 days of measurements a year. The NDACC sites which measure HCN vertical columns are distributed globally, with a higher concentration in the NH, especially in Europe and North America. Here we selected HCN ground-based measurements from four sites (Table 1), having a long and continuative measurements time series: Thule (Greenland) at NH high latitudes, Bremen (Germany) and Jungfraujoch (Switzerland) at NH mid-latitudes, and Mauna Loa (Hawaii, USA) in the tropics.

**Table 1.** Ground-based NDACC FTIR observation sites used in this study.

| Site | Coordinates | Altitude [m.a.s.l.] | Period |
| --- | --- | --- | --- |
| Thule, Greenland | 76.5° N, 68.7° W | 225 | 1999-present |
| Bremen, Germany | 53.1° N, 8.8° E | 27 | 2006-present |
| Jungfraujoch, Switzerland | 46.6° N, 8.0° E | 3580 | 1995-present |
| Mauna Loa, Hawaii (USA) | 19.5° N, 155.6° W | 3397 | 1995-present |

## 3 Model Simulation

We investigate HCN variability using an updated version of the TOMCAT three-dimensional (3-D) chemical transport model (CTM). TOMCAT is an Eulerian offline 3-D global CTM used for a wide range of tropospheric and stratospheric chemistry studies. It was originally developed by Chipperfield et al. (1993) as two different models used for tropospheric and stratospheric studies, respectively, called TOMCAT and SLIMCAT. The two models were subsequently combined into the TOMCAT/SLIM-CAT unified model (Chipperfield, 2006; Monks et al., 2017), which hereafter we call TOMCAT. TOMCAT has been used in a large number of studies and performs well in simulations of tropospheric (e.g Pope et al. (2020)) and stratospheric (e.g Chipperfield et al. (2018)) chemistry.

TOMCAT typically uses a flexible horizontal and vertical resolution with a $\sigma - p$ vertical coordinate system. The vertical
grid includes the surface $\sigma$ level which follows the terrain and pure pressure levels at higher altitudes up to 10 Pa (about 60
km). In the present study, the model was run at a spatial resolution of $2.8° \times 2.8°$ on a 60-level altitude grid from 2000 to 2016.
The meteorology of the model is forced by humidity, temperature and winds from ERA-Interim reanalyses provided by the
European Centre for Medium-Range Weather Forecasts (ECMWF) with a 6-hour time resolution (Berrisford et al., 2011; Dee
et al., 2011). The meteorological information is linearly interpolated to fit with the time step and the spatial grid chosen for
the model run. Natural and anthropogenic surface emissions are included in the model on a $1° \times 1°$ resolution and re-gridded
onto the model spatial grid. The HCN emissions are extracted from some principal emission datasets: the Coupled Model
Intercomparison Project Phase 6 (CMIP6) for anthropogenic and ocean emissions (Eyring et al., 2016), the Chemistry-Climate
Model Initiative (CCMI) for a fixed annual biogenic emission dataset (Morgenstern et al., 2017) and Global Fire Emissions
Database Version 4 (GFED4) for the biomass burning emissions (Randerson et al., 2017).

## 3.1 Upper Troposphere - Stratosphere HCN Loss

The TOMCAT simulation included four idealised HCN tracers (HCN1 to HCN4) to test the different atmospheric loss mech-
anisms of HCN. For each of these tracers, the global surface HCN volume mixing ratio (VMR) was constrained to a fixed
value of 200 ppt, approximately the background tropospheric VMR measured during the Transport and Chemical Evolution
Over the Pacific (TRACE-P) aircraft campaign and modelled by Li et al. (2003) and Singh et al. (2003). The HCN atmospheric
chemistry was modelled using the parameters summarized in Table 2. The main focus here is on the HCN removal process
via oxidation by OH radicals and the comparison of the two different reaction rates, the recommended rate proposed by JPL
(Burkholder et al., 2015, 2019) and the rate presented by Kleinböhl et al. (2006) based on the experimental measurements of
Strekowski (2001). The other loss process included in the four tracers are the HCN reaction with O($^1$D) (Kleinböhl et al., 2006;
Strekowski, 2001). The loss of HCN by photolysis is very slow and can be ignored (Burkholder et al., 2019).

Figure 3 compares the four model tracers with example average profiles measured by ACE-FTS averaged over 60 degree
latitude bands. Tracer HCN1 includes only the JPL recommended rate for HCN oxidation by reaction with OH. This tracer
substantially underestimates the amount of HCN in the stratosphere. The HCN1 profile shows a drastic reduction in the HCN
VMR at altitudes above $\sim 20$ km where the measured value is greater than 150 pptv. HCN2 is used to evaluate the HCN
oxidation by OH radicals using the Kleinböhl et al. (2006) rate constant. For this tracer the model VMRs are closer to the
measurements but, above 30 km at high latitudes in both hemispheres, the modelled HCN amount clearly overestimates the
ACE observations by about 60 pptv. Introducing the HCN destruction by O($^1$D) this gap between observations and model is
greatly reduced. Considering the O($^1$D) sink alone (tracer HCN3) or in combination with the HCN reaction with OH (tracer
HCN4), we obtain a much more reasonable agreement with the measured HCN profile in the mid-upper stratosphere. Tracers
HCN3 and HCN4 show that the stratospheric HCN loss is driven by the reaction with O($^1$D). This good agreement is further
support that loss of HCN by photolysis in the stratosphere is negligible.

**Table 2.** HCN atmospheric photochemical loss mechanisms used for the different TOMCAT model tracers.

| Reaction | Rate coefficient | Reference | Model tracer |
|---|---|---|---|
| $HCN + OH^a$ | $k_{OH} = 1.2 \cdot 10^{-13} \cdot \exp(-400/T_{prof})$ | Burkholder et al. (2015, 2019) | HCN1 |
| $HCN + OH^b$ | $k_{OH} = \frac{k_0[M] \cdot k_\infty}{k_0[M] + k_\infty} \cdot 0.8^{(1+(log_{10}(k_0[M]/k_\infty))^2)^{-1}}$ | Strekowski (2001); Kleinböhl et al. (2006) | HCN2, HCN4 |
| $HCN + O(^1D)$ | $k_{O1D} = 7.70 \cdot 10^{-13} \cdot \exp(100/T_{prof})$ | Strekowski (2001); Kleinböhl et al. (2006) | HCN3, HCN4 |
| $HCN + h\nu$ | | Burkholder et al. (2019) | Not included here |

[a] Rate coefficient is expressed in $\frac{cm^3}{molec \cdot s}$ and $T_{prof}$ is the temperature profile expressed in K.

[b] $k_0$ is given in $\frac{cm^6}{molec^2 \cdot s}$, $k_\infty$ in $\frac{cm^3}{molec \cdot s}$, [M] is the molecular air density.

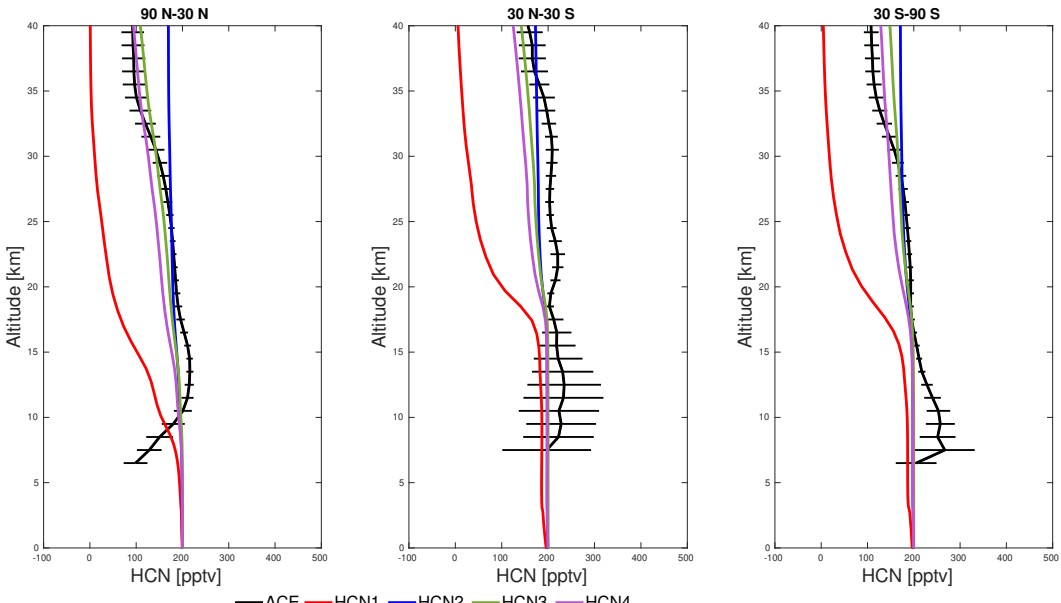

**Figure 3.** Average HCN profiles for March-May 2009 for 4 TOMCAT model tracers (HCN1- HCN4) compared with the profiles measured by ACE-FTS with ±1 median absolute deviation for latitude bands 90°N-30°N (left panel), 30°N-30°S (middle) and 30°S-90°S (right). See Table 2 for details of the photochemical loss reactions included in the different model HCN tracers.

## 3.2 Ocean Uptake

Li et al. (2000) and Li et al. (2003) both modelled HCN variability and developed two different schemes to reproduce the ocean uptake (Li et al., 2000, 2003). The first scheme was developed to test, using the 3-D model GEOS-Chem, the hypothesis that the biomass burning provides the main HCN source and the ocean uptake the main sink by focusing on the HCN seasonal features (Li et al., 2000). The Li et al. (2003) scheme was introduced into the GEOS-Chem model to perform new simulations with a longer HCN lifetime and weaker global sources than those of the previous study (Li et al., 2000), in order to match the TRACE-P constraints. The authors of Li et al. (2003) assert that their model achieved a similar or better simulation than in the study of Li et al. (2000).

In our study, a second set of six HCN tracers were implemented in TOMCAT to evaluate the two different ocean uptake schemes from Li et al. (2000) and Li et al. (2003). For these tracers the atmospheric photochemical loss was included as in tracer HCN4, with the HCN reactions with OH (using the rate constant from Kleinböhl et al. (2006)) and O($^1$D), as this tracer gave the best model representation of stratospheric HCN variability compared to ACE-FTS.

The ocean uptake flux of HCN proposed by Li et al. (2000) is defined as

$$F_g = k_w C_g K_H R T^a \qquad (\mathrm{kg\,m^{-2}s^{-1}}), \qquad (1)$$

**Table 3.** HCN ocean uptake rates used for the different TOMCAT model tracers.

| Ocean uptake flux $(\mathrm{kg\,m^{-2}s^{-1}})$ | Reference | Model tracer |
|---|---|---|
| $F_g = k_w C_g K_H R T^a$ | Li et al. (2000) | L2000 |
| $F_g = 0.5 \cdot k_w C_g K_H R T^a$ | Li et al. (2000) | L2000$_{0.5}$ |
| $F_g = 0.25 \cdot k_w C_g K_H R T^a$ | Li et al. (2000) | L2000$_{0.25}$ |
| $F_g = 0.0013 \cdot C_g^b$ | Li et al. (2003) | L2003 |
| $F_g = 2 \cdot 0.0013 \cdot C_g^b$ | Li et al. (2003) | L2003$_2$ |
| $F_g = 3 \cdot 0.0013 \cdot C_g^b$ | Li et al. (2003) | L2003$_3$ |

where $k_w = 0.31 u^2 (S_c/666)^{1/2}$ (m s$^{-1}$) is the air-to-sea transfer velocity with $u$ (m s$^{-1}$) the wind speed at 10 m, and the dimensionless parameter $S_c = \nu/D$ is the Schmidt number of HCN in water, with $\nu$ (m$^2$s$^{-1}$) the kinematic viscosity and $D$ (m$^2$s$^{-1}$) the diffusion coefficient of HCN in water. $C_g$ (kg m$^{-3}$) is the concentration of HCN in surface air, $K_H$ the temperature-dependent Henry's law constant defined as $K_H = 12$ M atm$^{-1}$ at 298K and $\Delta H_{298}/R = -5000$ K. $R = 287.05$ J kg$^{-1}$K$^{-1}$ is the gas constant, and $T$ (K) is the sea surface temperature.

The second ocean uptake scheme from Li et al. (2003) uses the results of a box model of the marine boundary layer (MBL) to derive an oceanic deposition velocity of 0.13 cm s$^{-1}$ (Singh et al., 2003). The resulting flux is defined as

$$F_g = 0.0013 \cdot C_g \qquad (\mathrm{kg\,m^{-2}s^{-1}}) \tag{2}$$

where $C_g$ (kg m$^{-3}$) is the HCN concentration near the surface.

The NDACC ground-based column measurements are used to evaluate the ocean uptake schemes, which act as the HCN surface sinks in the model, due to the large impact on the HCN budget and mean tropospheric VMR. The TOMCAT simulation is sampled at each NDACC station location and the profiles are smoothed using the instrument specific averaging kernels. The total column time series are then compared as shown in Figures S1-S4 in the Supporting Information. The agreement is evaluated considering the root mean square error (RMSE) and the coefficient of determination (R$^2$) values in reference to the measurements (Tables S1-S2 in the Supporting Information).

Tracers L2000 and L2003, which make use of the two published schemes from Li et al. (2000) and Li et al. (2003), respectively, show a good agreement with NDACC total column time series only in terms of the HCN seasonality. In fact, L2000 and L2003 show the highest RMSE values among all the tracers and a strongly negative R$^2$ at each location. The two tracers give greatly different results in our 3-D model compared to the HCN amount measured by FTIR instruments; L2000 underestimates the HCN total columns, the values are almost two-thirds of the observed values, while L2003 greatly overestimates them, the model is almost double the FTIR measured values. The observed mismatch is attributable to the ocean uptake fluxes being unable to capture contributions of this process to the HCN variability. We therefore need to scale the two sinks, in different directions, in order to reach a better representation of HCN variability. The new tracers L2000$_{0.5}$ and L2000$_{0.25}$ are created by

applying the scaling factors 0.5 and 0.25, respectively, to reduce the Li et al. (2000) ocean uptake flux while the tracers $L2003_2$ and $Li2000_3$, apply the factors 2 and 3, respectively, to the Li et al. (2003) flux in order to increase the HCN ocean uptake.

The tracers including the scaled ocean uptake schemes show a substantial improvement in the agreement with the NDACC measurements. In particular, the best agreement, considering the RMSE and $R^2$ values, is obtained by reducing the HCN flux in the Li et al. (2000) scheme by three quarters or by doubling the Li et al. (2003) flux in the $L2003_2$ tracer.

Figure 4 shows a comparison between the measured HCN total column time series and the two best-performing tracers $L2000_{0.25}$ and $L2003_2$, characterized by the lowest RMSE and a large $R^2$, and the worst one, L2003. HCN total columns

from the L2003 model tracer are larger than the HCN measured by FTIR instruments at all the locations, it is clearly visible especially for Jungfraujoch where the HCN estimated by L2003 is more than twice that the measured one. Both $L2000_{0.25}$ and $L2003_2$, the best-performing model tracers, agree very well with the measured values confirming that both ocean uptake schemes published by Li et al. (2000, 2003) need to be scaled to be more accurate in reproducing HCN variability.

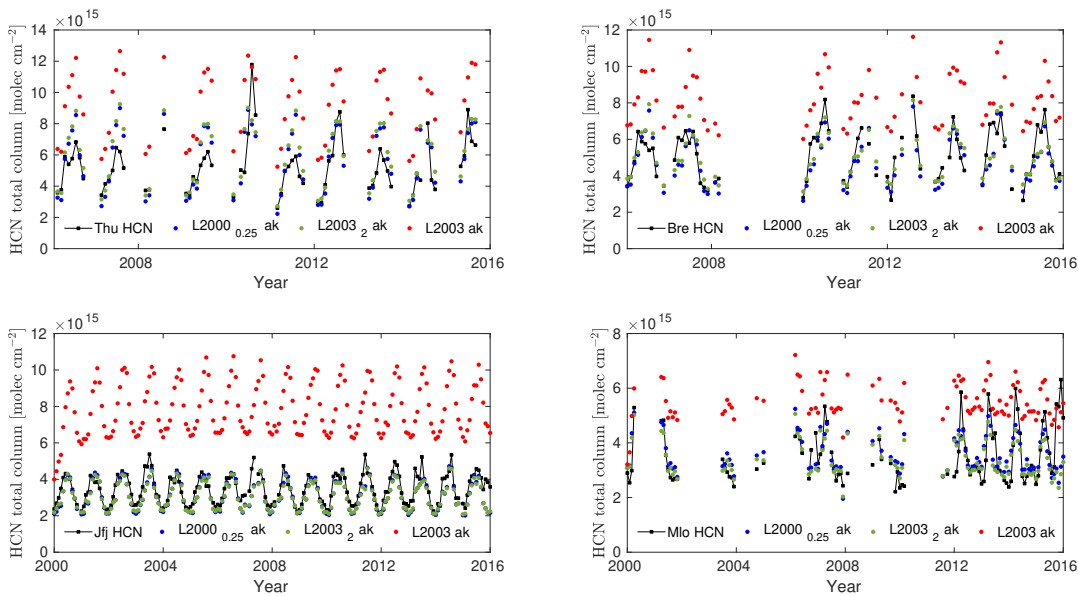

**Figure 4.** HCN total column time series (molecules $cm^{-2}$) measured at 4 NDACC stations (Thule (Thu), Bremen (Bre), Jungfraujoch (Jfj), and Mauna Loa (Mlo); black lines) and modelled by the $L2000_{0.25}$ (blue dots), $L2003_2$ (green dots) and L2003 (red dots) tracers (with averaging kernels applied). Note the different time periods at the different stations.

### 3.3 Atmospheric Lifetime and Global Budgets

The global budgets of atmospheric HCN for the model tracers L2000, L2003 and $L2000_{0.25}$ are shown in Table 4. Here we report only the two tracers using the original ocean uptake schemes from Li et al. (2000, 2003), which have the worst performance, and the best one using the ocean uptake from Li et al. (2000) scaled by 0.25. The total atmospheric burdens of

**Table 4.** Global burden, budget terms and atmospheric lifetimes for three model HCN tracers.

| Parameters | L2000 | L2003 | L2000$_{0.25}$ |
|---|---|---|---|
| Atmospheric Burden (Tg N) | 0.33 | 0.89 | 0.55 |
| Emissions (Tg N yr$^{-1}$) | 2.42 | 2.42 | 2.42 |
| Ocean Uptake (Tg N yr$^{-1}$) | 2.36 | 2.32 | 2.38 |
| Reaction with OH (Tg N yr$^{-1}$) | 0.05 | 0.12 | 0.12 |
| Reaction with O($^1$D) (Tg N yr$^{-1}$) | $7\times10^{-4}$ | $1.9\times10^{-3}$ | $1.9\times10^{-3}$ |
| Atmospheric lifetime (months) | 1.6 | 4.4 | 2.6 |

HCN are 0.33, 0.89 and 0.55 Tg N, respectively, reflecting the HCN concentration underestimation of the L2000 tracer and the overestimation of the L2003 tracer, while the burden of L2000$_{0.25}$ is close to the values reported in Li et al. (2000, 2003) and Singh et al. (2003). Each HCN tracer uses the same emission scheme, with biomass burning as the main contribution, producing 2.42 Tg N yr$^{-1}$. Ocean uptake provides the main sink for all the tracers, 2.36 Tg N yr$^{-1}$ in L2000, 2.32 Tg N yr$^{-1}$ in L2003 and 2.38 Tg N yr$^{-1}$ in L2000$_{0.25}$, which all largely balance the emissions despite the variation in the first-order rate of uptake (i.e. the change in atmospheric HCN burden compensates). The sink from reaction with OH is 0.05 Tg N yr$^{-1}$, 0.12 Tg N yr$^{-1}$ and 0.12 Tg N yr$^{-1}$ for L2000, L2003 and L2000$_{0.25}$, respectively. Despite its importance for determining the stratosphere loss of HCN, the reaction with O($^1$D) is a relatively very small sink globally, $7\times 10^{-4}$ for the L2000 tracer, $1.9\times 10^{-3}$ for L2003 and $1.9\times 10^{-3}$ Tg N yr$^{-1}$ for L2000$_{0.25}$. The resulting tropospheric lifetimes are 1.6, 4.4 and 2.6 months for the three tracers reported in Table 4.

The budgets of the L2000$_{0.25}$ tracer, which best reproduces the HCN variability, are in good agreement with the HCN budgets of Li et al. (2000), while the ocean uptake loss and the emissions are substantially larger than the Li et al. (2003) and the Singh et al. (2003) results. Li et al. (2003) estimates a global HCN loss to the ocean of 0.73 Tg N yr$^{-1}$ and Singh et al. (2003) of 1.0 Tg N yr$^{-1}$, both of these estimations are less than a half of the ocean uptake calculated in the present study. Similarly our emissions are more than twice the total emissions from Li et al. (2003), 0.63 Tg N yr$^{-1}$ from biomass burning and 0.2 Tg N yr$^{-1}$ from residential coal burning, and Singh et al. (2003), 1.1 Tg N yr$^{-1}$. The resulting global mean atmospheric lifetime is also in agreement with the range of 2.1-4.4 months calculated by Li et al. (2000).

## 4 Global Distribution of HCN

Figure 5 shows the simulated global mean distributions of HCN at the surface level for the six tracers created to test the ocean uptake schemes during September 2009. Higher surface concentrations of HCN (>1000 pptv) are observed over the biomass burning regions of South-East Asia, Central America, and Central Africa. The HCN surface concentrations are very low (<100 pptv) over the oceans, especially at high latitudes in the SH. This is due to the remote position from any HCN biomass burning emission sources, reflecting the role of the ocean uptake as the major removal mechanisms in the marine boundary layer. It

is important to highlight that the lack of ground-based observations in the SH is a limitation for better constraining the HCN distribution in this part of the world.

Figure 6 shows the seasonal latitude-height zonal mean cross sections for December 2008-November 2009 for the model tracers used to test the Li et al. (2000) and the Li et al. (2003) ocean uptake schemes. This figure reveals the presence of a generally asymmetric distribution of HCN between the two hemispheres, with a higher concentration of HCN in the NH, which has more biomass burning regions. As shown in Figure 1 and reported in the last row of Figure 6, HCN also has a strong seasonal pattern linked to the biomass burning emissions during the seasons of large wildfires. All the tracer cross sections generally reproduce this seasonal behaviour with some small differences. ACE-FTS data during season DJF exhibits a high HCN concentration in the upper troposphere over the SH between midlatitudes and high latitudes (Fig. 1). The model tracers highlight a band of high HCN concentrations over the equatorial region, with low concentrations at high latitudes in agreement with ACE-FTS observations. The tracers also agree well with the measurements in JJA, showing the HCN enhancement in the mid and upper troposphere from the NH mid- to high latitudes due to the wildfire season, with a peak in the southern equatorial regions near the surface level not observed by ACE-FTS measurements which only cover the altitudes above 5 km. During SON, the typical wildfire season in South America, Africa and South East Asia, the model tracers are also able to reproduce the HCN enhancement over the southern tropical region. During MAM all the tracers show an enhancement of HCN in the troposphere over the NH from the tropics to midlatitudes, reflecting the start of the wildfire season in the area, although in L2000 with its large ocean uptake this enhancement is extremely weak. In terms of the HCN amount, tracers L2000 and L2003, consistent with the comparison of TOMCAT tracers with ground-based measurements (Figures S1-S4), show low or extremely high HCN concentrations across the entire UTLS, respectively. As previously observed in the comparison between the HCN total column time series from the model tracers and the NDACC measurements, tracers $L2000_{0.25}$ and $L2003_2$ are the ones which show the best agreement, which is also the case for comparisons with ACE-FTS measurement.

## 5   Conclusions

We have presented HCN profiles from version 4.1 processing of the ACE-FTS data, and used these observations in the upper troposphere and stratosphere to evaluate different atmospheric HCN loss processes. The ACE-FTS observations extend to $\sim$42 km which is higher than typical balloon profiles which have previously been used to examine HCN chemistry. Using the ACE-FTS data we were able to test the processes driving HCN variability in the stratosphere through comparisons with a series of tracers in the TOMCAT 3-D model.

Our results confirm that the recommended rate from JPL (Burkholder et al., 2015, 2019) for the oxidation reaction between HCN and OH radicals greatly overestimates the HCN loss. The best agreement between the modelled and the measured profiles is obtained by using the reaction rate coefficient proposed by Kleinböhl et al. (2006) in combination with the HCN oxidation by $O(^1D)$. Loss via photolysis is assumed to be negligible in the altitude range considered (Burkholder et al., 2019) and can be ignored. Analysis of individual loss terms shows that reaction of HCN with $O(^1D)$ dominates in the mid-stratosphere. In

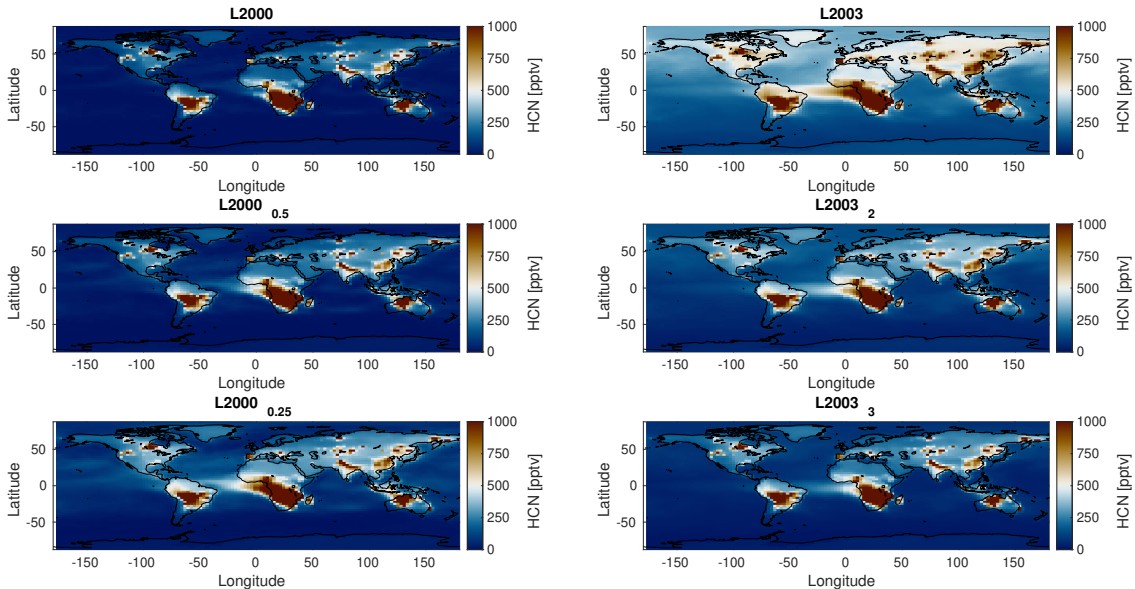

**Figure 5.** Simulated global distributions of HCN (pptv) for 6 model tracers L2000-L2003$_3$ at the surface level for September 2009.

particular, despite its small contribution to the overall atmospheric HCN budget, the reaction of HCN with O($^1$D) plays a significant role in determining the shape of the HCN profile in the stratosphere.

The major sink of atmospheric HCN is ocean uptake. However, the two published ocean uptake schemes of Li et al. (2000) and Li et al. (2003) in our 3-D model give greatly different results compared with NDACC data due to the inability of the schemes to correctly represent the HCN ocean uptake removal process. The tracer based on the Li et al. (2000) scheme (L2000) slightly underestimates the HCN concentration, while the tracer based on Li et al. (2003) (L2003) overestimates HCN. In order to obtain a more reasonable agreement, we scaled the ocean uptake fluxes. The best agreement was reached by reducing the Li

et al. (2000) flux by 75% or by doubling the Li et al. (2003) flux. In particular, the budgets of the tracer L2000$_{0.25}$ show a very good agreement with the previous studies, especially with Li et al. (2000).

     Overall, this work has demonstrated improvements in our ability to model the distribution of atmospheric HCN, an important marker of wildfire chemistry. The importance of HCN to track such events will likely increase in the future due to the projected increase in wildfires as a consequence of climate change (Abatzoglou et al., 2019; Jones et al., 2022; Senande-Rivera et al.,

2022). We have also demonstrated the role that satellite profiles can play in constraining the photochemical sinks of HCN. However, more work is need to better parameterise the ocean uptake which dominates the HCN budget globally.

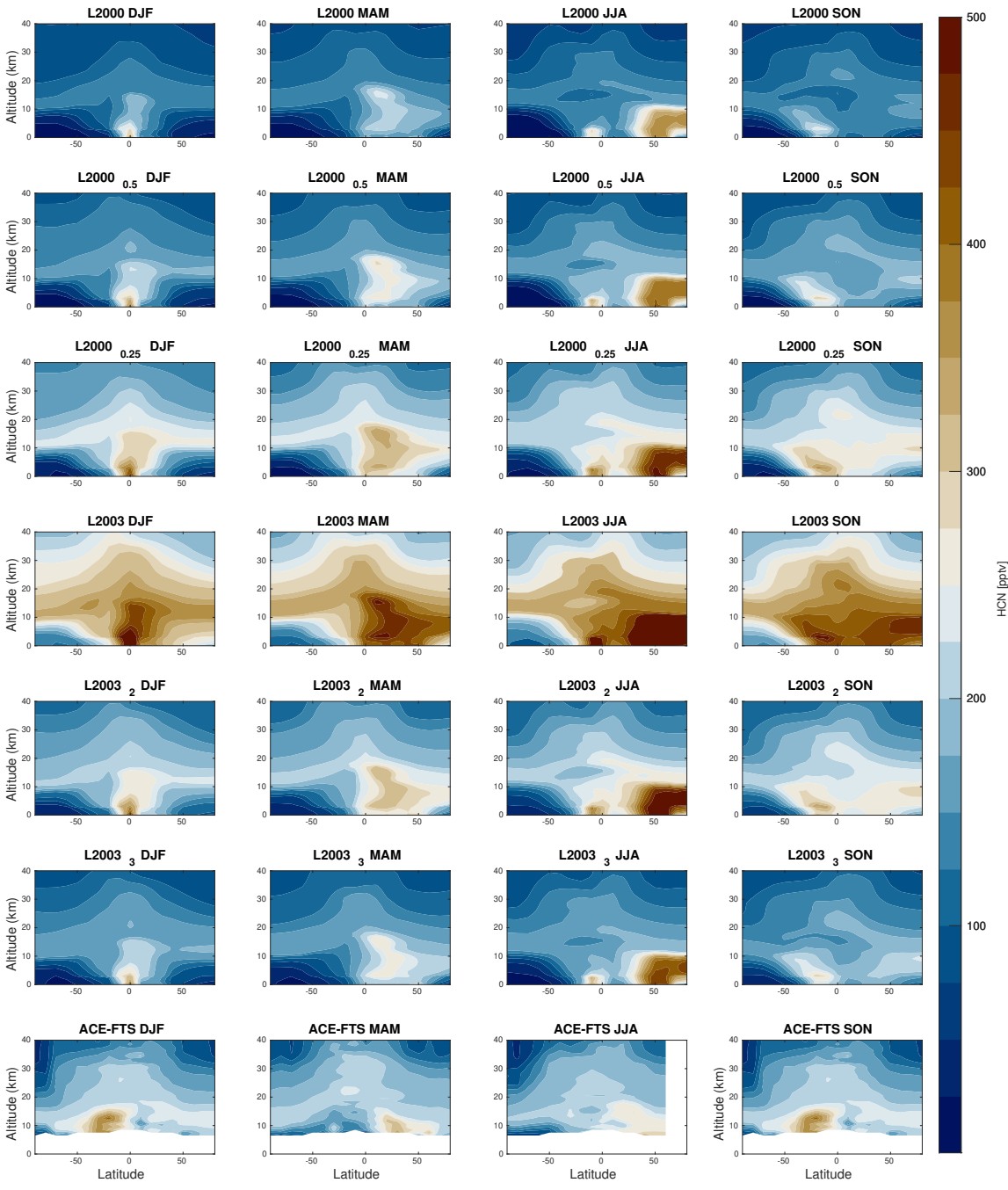

**Figure 6.** Seasonal mean latitude-height zonal mean cross sections from December 2008 to November 2009 of HCN zonal means in $10°$ latitude bins for 6 TOMCAT HCN tracers and ACE-FTS measurements (on the rows).

*Code and data availability.* ACE-FTS data were obtained from https://databace.scisat.ca/level2/ace_v4.1_v4.2/. NDACC data are publicly avilable at https://www-air.larc.nasa.gov/missions/ndacc/data.html. The TOMCAT model data are available at https://doi.org/10.5281/zenodo.

7228632

.

*Author contributions.* Model runs and data analysis were performed by AGB, with support from MPC and JJH. JJH and MPC designed the study. The TOMCAT model is maintained and updated by the MPC research group at the University of Leeds. Manuscript was written by AGB, with contributions from all the co-authors.

*Competing interests.* The authors declare that they have no conflicts of interest.

*Acknowledgements.* AGB was funded by a Natural Environment Research Council studentship at the University of Leicester, awarded through the Central England NERC Training Alliance (CENTA). This study was funded as part of the UK Research and Innovation Natural Environment Research Council's support of the National Centre for Earth Observation, contract number PR140015.

We thank R. Strekowski for helpful comments.

The model simulations were performed on the University of Leeds ARC HPC machines. The data analysis was performed on the AL-
ICE/SPECTRE High Performance Computing Facility (HPC) at the University of Leicester.

The ACE satellite mission is funded primarily by the Canadian Space Agency (CSA).

The FTIR data used in this publication were obtained from James Hannigan (Mauna Loa, Thule), E. Mahieu (Jungfraujoch), J. Notholt (Bremen) as part of the Network for the Detection of Atmospheric Composition Change (NDACC) and are available through the NDACC website www.ndacc.org.

The multi-decadal monitoring programme of ULiège at the Jungfraujoch station has been primarily supported by the F.R.S.-FNRS and by the GAW-CH programme of MeteoSwiss. The International Foundation High Altitude Research Stations Jungfraujoch and Gornergrat (HFSJG, Bern) supported the facilities needed to perform the FTIR observations.

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
