# Peer review of "Atmospheric Distribution of HCN from Satellite Observations and 3-D Model Simulations"

_EGUsphere, 2022_

## Referee Comment (RC1)

**Review of "Atmospheric Distribution of HCN from Satellite Observations and 3-D Model Simulations" by Bruno et al."**

Hugh C. Pumphrey

December 19, 2022

**General remarks**

This paper compares some new model simulations of HCN in the atmosphere to the solar occultation data from the ACE-FTS instrument. It is good to see some modelling done on this species because there remain a number of significant uncertainties in its sources and sinks. As the authors make clear, these include the rate constant for an important reaction and, more seriously, the rate at which HCN is lost from the atmosphere to the oceans.

The paper is well-written and easy to read; I have few corrections to make to the text. The figures are generally well designed and clear, except for the use of the much-derided "jet" colour scale. See `https://doi.org/10.1038/s41467-020-19160-7` for a recent discussion of colour scales, including details on why "jet" should be avoided, and a better colour scale called "batlow". If the authors would like a scale with similar colours to jet they might try Google's "turbo" scale: see `https://ai.googleblog.com/2019/08/turbo-improved-rainbow-colormap-for.html`.

The text labels on the figures often appears a bit small. It is difficult to judge this from the preprint; the authors should ensure that in the final version the text on figures is no smaller than the text in the caption.

My recommendation is that the paper should be accepted with minor revisions. My main non-cosmetic concern is whether the DJF panel of Figure 1 is correct.

**Specific corrections**

- Page 4, Figure 1: The DJF panel looks very different from the other three seasons, through the entire stratosphere. Is this real? Is it an artefact of the ACE-FTS sampling pattern? The difference appears to be such that, at $30\,\mathrm{km}$, there is about $200\,\mathrm{pptv}$ for most of the year, but $280\,\mathrm{pptv}$ in DJF. Is the same pattern observed in other years? I note that the time series in Figure 2 does not make it seem that DJF differs from the other seasons. The six modelled tracers shown in Figure 6 all have a DJF season which is generally similar to the other seasons. For interest I show the equivalent plots for the MLS data in Figure 1; the DJF season does not appear clearly different from the other seasons in the way that it does in the reviewed paper.

[Figure]

Figure 1: MLS V5 seasonal means of HCN volume mixing ratio (in pptv) for the four seasons of the 2008-9 year, as per Figure 1 of the paper reviewed. Note that the DJF season does appear to have slightly higher mixing ratios in the 30–40 km altitude range, but nowhere near to the extent suggested by Figure 1 of the reviewed paper. The systematic errors below 24 km which were the reason for the reviewed paper not using the MLS data are clear, especially at high latitudes. The colour map used is Google's "turbo".

- Page 4, Figure 1, Page 13, Figure 6 and any other filled contour plots: The colour bar should have bands of colour which match exactly with the colours used in the contour plots. It should not have a continuous graduation of colour from one end to the other.

- Page 5, Figure 2. On this figure, some of the text appears too large. I would also suggest that the entire figure is made taller, so that the altitude scale is less compressed. (As this figure is not a contour plot, but is an image with large pixels, a continuous colour bar is appropriate.)

- Page 6, lines 136–139: At this point there should be reference callouts to the three sources of emissions data. Datasets should be referenced in the same way as articles, and should also be mentioned in the "Code and data availability" section. Where a dataset has a DOI, it should be used. For example, the DOI for GFED4 is 10.3334/ORNLDAAC/1293

- Page 8, Figure 3: The axis labels on this figure are VERY small. The authors should consider the choice of colours for the five lines. Most seriously, pure

yellow should never be used for lines on plots as it is almost invisible on a white background. Less seriously, to my eyes the red (HCN1) and magenta (HCN4) lines are not easy to distinguish.

- Page 13, Figure 6: it might be worth adding an extra column to this figure to show the ACE-FTS data from Figure 1, for easier comparison. If this is done it would probably be necessary to eliminate one of the TOMCAT tracers to ensure that the individual plots were a reasonable size. One could have the season increase horizontally and the tracer name change vertically; this would allow the individual plots to be a bit larger. The colour bar should be divided into bands which match the contour levels used in the plots, as I said for Figure 1.

**Technical corrections**

- Page 9, Lines 175–177: Units should not be in italics. Also, it is preferable to use negative powers in units rather than the / symbol. For example, write $\mathrm{m\,s^{-1}}$ rather than m/s.

- Page 11 Figure 4: Horizontal axis labels should probably specify that time is in years (CE). The values in the range 2000 to 2016 make it unlikely that the labels could mean anything else, but one should not leave room for doubt. The authors might like to consider the choice of colours: on my screen it is quite hard for the eye to separate the black and blue dots.

---

## Author Comment (AC1)

<h1 style="text-align:center">Response to Reviewer's Comments</h1>

We thank the reviewers for taking the time to evaluate our manuscript and for their positive and helpful comments. These comments are reproduced below in *italics*, followed by '>>' and our responses.

**Reviewer 1**

*This paper compares some new model simulations of HCN in the atmosphere to the solar occultation data from the ACE-FTS instrument. It is good to see some modelling done on this species because there remain a number of significant uncertainties in its sources and sinks. As the authors make clear, these include the rate constant for an important reaction and, more seriously, the rate at which HCN is lost from the atmosphere to the oceans.*

*The paper is well-written and easy to read; I have few corrections to make to the text.*

**Specific Corrections**

*1) Page 4, Figure 1: The DJF panel looks very different from the other three seasons, through the entire stratosphere. Is this real? Is it an artefact of the ACE-FTS sampling pattern? The difference appears to be such that, at 30 km, there is about 200 pptv for most of the year, but 280 pptv in DJF. Is the same pattern observed in other years? I note that the time series in Figure 2 does not make it seem that DJF differs from the other seasons. The six modelled tracers shown in Figure 6 all have a DJF season which is generally similar to the other seasons. For interest I show the equivalent plots for the MLS data in Figure 1; the DJF season does not appear clearly different from the other seasons in the way that it does in the reviewed paper.*

>> We changed the "jet" colour scale to one which is more accessible for people with colour-vision deficiencies. The colour scale now used is the "vik" divergent scale from the Crameri library, for which there is more information at https://www.fabiocrameri.ch/colourmaps/.

The DJF panel looked different because it wrongly used a different colour scale (0-400 pptv instead of 0-500 pptv). The error has been fixed in the revised paper.

*2) Page 4, Figure 1, Page 13, Figure 6 and any other filled contour plots: The colour bar should have bands of colour which match exactly with the colours used in the contour plots. It should not have a continuous graduation of colour from one end to the other.*

>> In the revised paper we use a discrete colour bar with a lower number of bins showing exactly the same colours used in the figure.

*3) Page 5, Figure 2. On this figure, some of the text appears too large. I would also suggest that the entire figure is made taller, so that the altitude scale is less compressed. (As this figure is not a contour plot, but is an image with large pixels, a continuous colour bar is appropriate.)*

>> We have made the figure taller and reduced the size of the text on the axis labels.

*4) Page 6, lines 136{139: At this point there should be reference callouts to the three sources of emissions data. Datasets should be referenced in the same way as articles, and should also be mentioned in the "Code and data availability" section. Where a dataset has a DOI, it should be used. For example, the DOI for GFED4 is 10.3334/ORNLDAAC/1293*

>> We added references to the datasets in the text.

5) *Page 8, Figure 3: The axis labels on this figure are VERY small. The authors should consider the choice of colours for the five lines. Most seriously, pure yellow should never be used for lines on plots as it is almost invisible on a white background. Less seriously, to my eyes the red (HCN1) and magenta (HCN4) lines are not easy to distinguish*.

>> The font size of the labels has been increased and the line colours have been changed to colours that are more accessible.

6) *Page 13, Figure 6: it might be worth adding an extra column to this figure to show the ACE-FTS data from Figure 1, for easier comparison. If this is done it would probably be necessary to eliminate one of the TOMCAT tracers to ensure that the individual plots were a reasonable size. One could have the season increase horizontally and the tracer name change vertically; this would allow the individual plots to be a bit larger. The colour bar should be divided into bands which match the contour levels used in the plots, as I said for Figure 1*.

>> We edited the figure following the suggestion. We changed the orientation of the plots, the season changes horizontally and the tracer vertically, at the bottom of the figure were also added the ACE-FTS data for comparison. The colour bar has been changed showing exactly the same colour used in the figure (see the Comment #2).

**Technical corrections**

7) *Page 9, Lines 175-177: Units should not be in italics. Also, it is preferable to use negative powers in units rather than the / symbol. For example, write $ms^{-1}$ rather than m/s.*

>> The units are now reported in this form in the revised paper.

8) *Page 11 Figure 4: Horizontal axis labels should probably specify that time is in years (CE). The values in the range 2000 to 2016 make it unlikely that the labels could mean anything else, but one should not leave room for doubt. The authors might like to consider the choice of colours: on my screen it is quite hard for the eye to separate the black and blue dots*.

>> The x label "Time" has been replaced with "Years" in the revised paper. We also replaced magenta dots with a more accessible colour and we used a lighter blue to help to distinguish the blue and black dots.

**Reviewer 2**

*This study presents much improved understanding of the atmospheric processes that are controlling the HCN budget using the observational data sets and the global chemical transport model simulations. There are big uncertainties in the atmospheric loss processes and ocean update of HCN and this study certainly is valuable in better representation of those processes. The use of both the NDACC and the ACE-FTS satellite measurements supports the robustness of the comparison. In terms of methodology, I have little to suggest for improvements. However, I personally think that by adding a little bit more scientific background this paper will become more interesting to a broader community. Below are my comments for the authors may take into consideration.*

**Major Comments**

1) *I would like to see clearer description of what we do and do not understand about the HCN budget in introduction. All the references included in introduction seem to suggest we have good understanding about HCN in the troposphere and stratosphere and the last sentence seems to suggest otherwise. For instance, including a statement saying all the model simulations show different results would help supporting this argument.*

>> Despite it being well known that the HCN is emitted predominantly during biomass burning, the processes driving its variability in the atmosphere and their relative contributions are still not well understood, in particular the rate constant for the HCN reduction by reaction with OH radicals and the rate of the HCN loss by ocean uptake.

Several studies have investigated the HCN ocean uptake presenting different approaches, a number of reaction rates have been used to model the HCN reduction by ocean uptake. This uncertainty on contribution of the different process on HCN variability causes large variations in estimating the HCN budgets.

We have tried to clarify the above points in the revised paper. For example, we added the sentence "The rate constant for the reaction between HCN and OH and the rate at which HCN is reduced by ocean uptake still have a number of significant uncertainties." at the end of the first paragraph in Section 1. There are various other additions made in response to the other comments.

2) *The motivation of this study has to be mentioned clearly. Both observational data and the model simulations are used as tools but it is not clear what the main goal of this study is.*

>> We have added some sentences in the Introduction to make clear the main goals of our study.

3) *It is a reasonable approach to compare the model outputs to the NDACC in-situ. However, can we trust the ACE-FTS measurements quantitatively? Are there any in-situ measurements of HCN in the upper troposphere and lower stratosphere to confirm that the ACE-FTS measurements are reasonable?*

>> ACE-FTS profiles are widely used as reference for interpreting ground-based and nadir satellite measurements of HCN and to help validate tropospheric-stratospheric transport in atmospheric models (Park et al. 2013; Viatte et al. 2014; Glatthor et al. 2015; Sheese, et al. 2017). This information has been mentioned in the first paragraph of Section 2.1. Note that the ground-based NDACC data are remote (using the technique described) and not in-situ.

**Minor Comments:**

1) *L2-22 (Abstract): I would recommend moving some of the contents explaining the background to introduction, which would make abstract a little more compact. Also put emphasis on the key findings and significance of this study. For instance, why is simulating the HCN amount using the global model correctly important?*

>> We have removed some text from the abstract and also added some to better explain the motivation and significance.

2) *L2: It is mentioned that the physical and chemical processes are investigated. Are the HCN removal processes considered to be physical?*

>> The ocean uptake is considered as a physical removal process, in contrast to the removal by chemical reactions in the atmosphere.

3) *L6-9: I would recommend combining these into one simple sentence. My recommendation for a replacement would be, 'We detected the changes in the atmospheric composition due to large wildfire events over Indonesia in 2006 and 2015 using long-term measurements from the ACE-FTS'.*

>> OK. We have combined them into the sentence: "The long-term ACE-FTS measurements reveal the strong enhancements in upper tropospheric HCN due to large wildfire events in Indonesia in 2006 and 2015."

4) *L10: 'previous lower altitude balloon comparisons' can be removed.*

>> We think an important advance of the ACE comparisons is that they extend higher than the balloon data. Therefore we can examine chemical loss in the middle stratosphere. We think the comparison of the altitudes is important.

5) *L49: A citation for NDACC could be included here.*

>> Done. We have added De Mazière et al. 2018.

6) *L 67: I think this is a good place to make a statement of why a global transport model is being used to understand specific aspects of the HCN budget. Is TOMCAT model proved to be a good tool for this study?*

>> We have added a statement at the start of this paragraph. Yes, we believe that TOMCAT, as a state-of-the-art CTM, is a good tool for this. We have added further model references in Section 3.

7) *L70: Do Burkholder et al. (2015 & 2019) papers have the same rate coefficient for HCN oxidation?*

>> Yes, the reported rate for OH+HCN reported is the same in the two reports. In fact, the JPL rate was last evaluated in 1983. This has been noted.

8) *L83: This sentence can be augmented or replaced by a statement of the ACE-FTS sampling pattern being densely located in high latitudes.*

>> This information has been added.

9) *P4, Fig. 1: A couple of solid contours can be added to show the distributions in the stratosphere more clearly. The current color scheme makes it hard to distinguish the blue and the green area.*

>> The colour scale has been changed to a more accessible one (See Reviewer #1 comments). To help the reader we have added the location of the tropopause based on the ECMWF reanalyses (Maddox and Mullendore 2018). Details are given in the figure caption and in Section 2.1.

10) *L92: A reason why only 2008-2009 are used in the seasonal mean could be given here. Have there been any studies showing the HCN climatology in the past? Is ACE-FTS climatology reasonable compare to other data set quantitatively?*

>> We have reported 2008-2009 as a sample year for the entire timeseries reported in Figure 2 in time-altitude section for three different latitude section. We have clarified this in the text. Several studies have shown the HCN climatology, such as Park et al. 2013; 2021; Glatthor et al. 2015 and Koo et al. 2017. These

studies found a good agreement between the ACE-FTS climatology and the HCN distribution measured by other instruments, such as MLS, MIPAS and airborne missions.

11) *P5, Fig. 2: The color scheme can be revisited to show the variability in the stratosphere clearly. Also, what is the maximum value in 2015? The color bar Is saturated at 600 pptv.*

>> The colour scale has been changed using a more accessible one (See Reviewer #1 comments). The colour bar maximum value has been chosen to maximize the visibility of the atmospheric HCN concentration in all the three panels. The saturated values in the plume in late 2015 in the middle panel reached values larger than 1300 pptv. This has been noted in the caption.

12) *L133: What is the sampling frequency of HCN through the NDACC?*

>> We now better describe the sampling frequency of NDACC FTIR spectrometer in the first paragraph of Section 2.2.

13) *L120: with a higher concentration -> with higher concentrations*

>> We disagree with this suggestion. The text refers to the number of measuring sites so 'a higher concentration' seems ok to us.

14) *L134: A citation is needed for the ECMWF meteorological inputs. Also, does TOMCAT reproduce reasonable climatology of other tracers, such as carbon monoxide?*

>> We have now included as references for ERA-Int the following two papers: Berrisford et al. 2011 and Dee et al. 2011. Yes, TOMCAT does reasonable well (as other CTMs) in simulation CO (e.g.Monks, 2012; 2017).

15) *P8, Fig. 3: The yellow lines are almost invisible on screen. A replacement color is recommended.*

>> Yellow and magenta lines have been replaced using more accessible colours. See Reviewer #1 comments.

16) *P14 (Conclusion): I would recommend adding a statement what role HCN is playing in our current climate and how this study will contribute predicting future climate better related to the results presented in this work.*

>> We do not fully understand this comment. HCN is an important tracer of biomass burning and can be considered to lead to poor air quality. It is not normally considered as a 'climate gas'. One link to climate is that HCN emissions may increase due to increased occurrence of wildfires in the future. We have added some text at the end of the Conclusions section to note this.

Berrisford, P., D. P. Dee, P. Poli, R. Brugge, Mark Fielding, Manuel Fuentes, P. W. Kållberg, S. Kobayashi, S. Uppala, and Adrian Simmons. 2011. *The ERA-Interim Archive Version 2.0*. *ERA Report Series*. 1. Shinfield Park, Reading: ECMWF.
De Mazière, Martine, Anne M. Thompson, Michael J. Kurylo, Jeannette D. Wild, Germar Bernhard, Thomas Blumenstock, Geir O. Braathen, et al. 2018. 'The Network for the Detection of Atmospheric

Composition Change (NDACC): History, Status and Perspectives'. *Atmospheric Chemistry and Physics* 18 (7): 4935–64. https://doi.org/10.5194/acp-18-4935-2018.

Dee, D. P., S. M. Uppala, A. J. Simmons, P. Berrisford, P. Poli, S. Kobayashi, U. Andrae, et al. 2011. 'The ERA-Interim Reanalysis: Configuration and Performance of the Data Assimilation System'. *Quarterly Journal of the Royal Meteorological Society* 137 (656): 553–97. https://doi.org/10.1002/qj.828.

Glatthor, N., M. Höpfner, G. P. Stiller, T. von Clarmann, B. Funke, S. Lossow, E. Eckert, et al. 2015. 'Seasonal and Interannual Variations in HCN Amounts in the Upper Troposphere and Lower Stratosphere Observed by MIPAS'. *Atmospheric Chemistry and Physics* 15 (2): 563–82. https://doi.org/10.5194/acp-15-563-2015.

Koo, Ja-Ho, Kaley A. Walker, Ashley Jones, Patrick E. Sheese, Chris D. Boone, Peter F. Bernath, and Gloria L. Manney. 2017. 'Global Climatology Based on the ACE-FTS Version 3.5 Dataset: Addition of Mesospheric Levels and Carbon-Containing Species in the UTLS'. *Journal of Quantitative Spectroscopy and Radiative Transfer*, Satellite Remote Sensing and Spectroscopy: Joint ACE-Odin Meeting, October 2015, 186 (January): 52–62. https://doi.org/10.1016/j.jqsrt.2016.07.003.

Maddox, Emily M., and Gretchen L. Mullendore. 2018. 'Determination of Best Tropopause Definition for Convective Transport Studies'. *Journal of the Atmospheric Sciences* 75 (10): 3433–46. https://doi.org/10.1175/JAS-D-18-0032.1.

Monks, S. A., S. R. Arnold, and M. P. Chipperfield. 2012. 'Evidence for El Niño–Southern Oscillation (ENSO) Influence on Arctic CO Interannual Variability through Biomass Burning Emissions'. *Geophysical Research Letters* 39 (14). https://doi.org/10.1029/2012GL052512.

Monks, Sarah A., Stephen R. Arnold, Michael J. Hollaway, Richard J. Pope, Chris Wilson, Wuhu Feng, Kathryn M. Emmerson, et al. 2017. 'The TOMCAT Global Chemical Transport Model v1.6: Description of Chemical Mechanism and Model Evaluation'. *Geoscientific Model Development* 10 (8): 3025–57. https://doi.org/10.5194/gmd-10-3025-2017.

Park, Mijeong, William J. Randel, Douglas E. Kinnison, Louisa K. Emmons, Peter F. Bernath, Kaley A. Walker, Chris D. Boone, and Nathaniel J. Livesey. 2013. 'Hydrocarbons in the Upper Troposphere and Lower Stratosphere Observed from ACE-FTS and Comparisons with WACCM'. *Journal of Geophysical Research: Atmospheres* 118 (4): 1964–80. https://doi.org/10.1029/2012JD018327.

Park, Mijeong, Helen M. Worden, Douglas E. Kinnison, Benjamin Gaubert, Simone Tilmes, Louisa K. Emmons, Michelle L. Santee, Lucien Froidevaux, and Chris D. Boone. 2021. 'Fate of Pollution Emitted During the 2015 Indonesian Fire Season'. *Journal of Geophysical Research: Atmospheres* 126 (9): e2020JD033474. https://doi.org/10.1029/2020JD033474.

Sheese, Patrick E., Kaley A. Walker, and Chris D. Boone. 2017. 'A Global Enhancement of Hydrogen Cyanide in the Lower Stratosphere throughout 2016'. *Geophysical Research Letters* 44 (11): 5791–97. https://doi.org/10.1002/2017GL073519.

Viatte, C., K. Strong, K. A. Walker, and J. R. Drummond. 2014. 'Five Years of CO, HCN, $C_2H_6$, $C_2H_2$, $CH_3OH$, HCOOH and $H_2CO$ Total Columns Measured in the Canadian High Arctic'. *Atmospheric Measurement Techniques* 7 (6): 1547–70. https://doi.org/10.5194/amt-7-1547-2014.